# Transferred Bacterial Community on the Potentially Pathogenic Bacteria among Aquatic Water, Plant Root, and Sediment When Planting with Chinese Herbs

**Yao Zheng** [1,2] , **Jiawen Hu** [1] **and Gangchun Xu** [1,2,*]

[1] Key Laboratory of Integrated Rice-Fish Farming Ecology, Ministry of Agriculture and Rural Affairs, Freshwater Fisheries Research Center (FFRC), Chinese Academy of Fishery Sciences (CAFS), No. 9 Shanshui East Road, Wuxi 214081, China; zhengy@ffrc.cn (Y.Z.); hujw@ffrc.cn (J.H.)
[2] Wuxi Fishery College, Nanjing Agricultural University, No. 9 Shanshui East Road, Wuxi 214081, China
[*] Correspondence: xugangchun1979@163.com; Tel.: +86-510-85556266

**Abstract:** With the development of modern aquaculture, the number of pathogenic bacteria in fish farms has gradually risen. Studies have shown that traditional Chinese medicinal herbs and natural products have greatly contributed to reducing bacterial growth and reproduction. To explore the changes in different proportions of *Houttuynia cordata* Thunb and *Jussiaea stipulacea* on the bacterial composition in water, roots, and sediments, we conducted 16S rRNA gene sequencing on samples of the same to analyze floating beds (60% *H. cordata* Thunb and 30% *H. cordata* Thunb, 30% *J. stipulacea* named HcT1, HcT2, and Jr, respectively) after 30 days in the presence of tilapia culture water, roots, and sediments with bacterial community changes in the respective experimental groups. The results showed that 4811 bacterial operational taxonomic units (OTUs) were obtained; the alterations included decreased Spirochaetae, Nitrospirae, and Elusimicrobia in water; a significant increase in Tenericutes, Chlorobi, and Nitrospirae in HcT1 roots; and decreased Firmicutes and Fusobacteria in HcT2 and Jr roots. Actinobacteria, Nitrospirae, Tenericutes, and Chlamydiae increased in the HcT1 sediment; Fusobacteria and Fibrobacteres increased in the HcT2 sediment; and Cyanobacteria, Gemmatimonadetes, and Acidobacteria increased in the Jr sediment. *H. cordata* Thunb decreased Tenericutes and Deferribacteres, while Chlorobi, Nitrospirae, and Gemmatimonadetes increased with a 60% planting area, whereas Actinobacteria and Cyanobacteria increased with a 30% planting area, and Jr only increased Fusobacteria and Fibrobacteres. When planting with herbs, Proteobacteria increased, while Deferribacteres and Elusimicrobia decreased. The pathogenic genera may transfer among the water, plant roots, and sediments, and floating cultivation with herbs may be beneficial for blocking the spread of the pathogenic genera found in the samples.

**Keywords:** Chinese herbs; *Houttuynia cordata* Thunb; *Jussiaea stipulacea*; sediment; Nitrospirae

## 1. Introduction

Common pathogenic bacteria in aquaculture water include *Aeromonas hydrophila*, *Vibrio parahaemolyticus*, *Bacillus cereus*, *Pseudomonas fluorescens*, and *Citrobacter freundii*. Congestion and ulceration of the surface of prawns and marine fish are caused by the hemolytic toxin secreted by *V. parahaemolyticus*, while *P. fluorescens* and *C. freundii* are conditional pathogens that can cause animal sepsis leading to acute death [1]. At present, the most commonly used probiotics in feeds at fish farms are *Bacillus*, *Lactobacillus*, *Lactococcus*, and *Saccharomyces*. The study of traditional Chinese medicinal herbs and natural products has greatly contributed to the prevention and treatment of illnesses [2,3]. Our previous studies showed that *Houttuynia cordata* Thunb floating beds can significantly improve non-specific immunity in tilapia [3]. *Jussiaea stipulacea* can eliminate algae by competing with cyanobacteria for light, nitrogen (N), phosphorus (P), and other nutrients [4]. Floating bed cultivation with *J. stipulacea* released eight bioactive compounds (palmitic acid, sitosterol,

betulinic acid, gallic acid, ursolic acid, quercetin 3-O-rhamnoside, kaempferol, and luteolin), which played the role of allelochemicals for harmful bacterial prevention. A previous study revealed that volatile allelochemicals had a weak antioxidative ability [5], and berberine (a kind of alkaloid extracted from plants) has been found as the main allelochemical from the root of *Coptis chinensis*, implementing the disease inhibitory effects [6]. However, some vegetable-like plants, like water spinach (*Ipomoea aquatica* forsk), which can be useful for in-situ remediation with the enhancement of water quality and the removal of pesticide simultaneously, can absorb methomyl [7], and such aquatic plants have the advantage of nutrient absorption [7]. Our recent study demonstrated that planting with mint (*Mentha haplocalyx* Briq) can alleviate the toxicological effects produced by methomyl, shape the intestinal microbiota in tilapia, and strengthen the connection between pond water quality and fish metabolic parameters [8]. For this reason, there has been considerable interest in the use of medicinal herbs in aquaculture to provide safe and eco-friendly replacements for antibiotics and chemical compounds, as well as to enhance immunity and prevent fish diseases [2,3].

Crosstalk exists between the plant root, water, and sediment in the pond [9]. Pathogenic bacteria could be isolated from the pond water and its corresponding plant roots [10]. Non-tuberculous mycobacteria (NTM) have been found in water, sediment, and aquatic plant samples [11]. Virulent *Aeromonas hydrophila* (vAh) was detected in 35.4% and 22.9% of biofilm and sediment samples, respectively [12]. The presence of NTM in sediments and aquatic plants in fishponds confirmed the possible transmission of mycobacteria from the aquatic environment into the fish [13]. Genera associated with pathogens, such as Acinetobacter, Arcobacter, and Clostridium, were also observed in water and sediment samples [14]. Pathogenic prevention by traditional Chinese medicinal herbs has been reported in *Polygonum cuspidatum, H. cordata* Thunb, *J. stipulacea*, and other herbs, which showed that the ratio of Bacteroidetes to Firmicutes was significantly increased, and in the algae (i.e., Anabaena and Microcystis), beneficial pathogenic bacteria increased and decreased for enhancing immune activity [2,3]. The hypothesis for this study was that the pathogenic genera may transfer among the water, plant roots, and sediments, and floating cultivation with herbs may be beneficial for blocking the spread of the pathogenic genera found in the samples.

## 2. Materials and Methods

### 2.1. Experiment Design and Sample Collection

The floating bed design was described [2,3], proposing combined meshes with a plastic pipe (spacing 20 × 30 cm) loaded with different plants using a rope to bind with the glass tank. One-fourth of the glass tanks (500 L) were divided into the control (CK as the abbreviation for this group, without floating bed cultivation, Figure 1), while 60% (HcT1 as the abbreviation for this group) and 30% (HcT2 as the abbreviation for this group) planting areas were cultivated with *H. cordata* Thunb and *J. stipulacea* (Jr as the abbreviation for this group, 30% planting area), respectively ($n$ = 3 glass tanks; in triplicates). All sequence samples from the four different plant groups were collected on days 0 and 30, each from June to July 2021. In each tank per sampling point, samples were sequenced for a high-throughput sequencing analysis ($n$ = 3; three samples were mixed in each tank to make up one sequencing sample), and preprocessing, and sequencing methods followed our previous study [2,3]. Tilapia (24.56 ± 3.14 g, $n$ = 16 per tank) was obtained from the Yixing research center of FFRC-CAFS, and the culture conditions maintained without water change and oxygenation and feeding followed the previous study [2,3].

### 2.2. DNA Extraction, PCR Amplification of 16S rDNA, Amplicon Sequencing, and Sequence Data Processing

Total bacterial DNA extraction from the samples followed by PCR amplification was performed as described in [2]. The barcoded primers for the PCR amplification of bacterial 16S rRNA genes of V3–V4 were 343F-5′-TACGGRAGGCAGCAG-3′ and 798R-

5′-AGGGTATCTAATCCT-3′ [2]. Finally, all PCR products were quantified using a Qubit dsDNA assay kit (invitrogen) and pooled together. A high-throughput sequencing analysis of bacterial rRNA genes was performed on the purified, pooled sample using the Illumina HiSeq 2500 platform (2 × 250 paired ends) at the OE Biotech Corporation (Shanghai, China) [2]. The filtration, cleaning, taxonomy annotation, core microbiome screening, and calculation of bacterial community indices (Chao1, Ace, Shannon, and Simpson) were carried out [2,3].

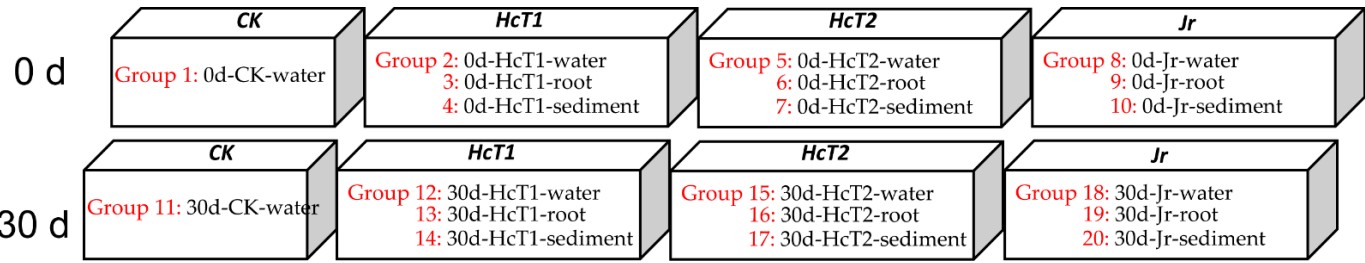

**Figure 1.** The experimental design containing the sample names. Group 1 comprised only water as a control (named as 0d-CK-water); Group 2 comprised water samples with HcT1 (0d-HcT1-water); Group 3 comprised root samples with HcT1 (0d-HcT1-root); and Group 4 comprised sediment samples with HcT1 (0d-HcT1-sediment). The labeling of groups 5–20 was ordered according to the exposure treatments.

*2.3. The Possible Transformation and Beneficial Influence for Prevention of Pathogenic Bacteria*

We applied a previously used relative abundance threshold (0.5%) [15] to focus our analysis on PCR-reproducible operational taxonomic units (OTUs). We selected significant increases or decreases in the bacterial community to obtain the change from the databank constructed using a set of comparisons, and the relative enriched KEGG pathways were simultaneously analyzed [2–4,13], e.g., comparisons from different planting species: groups 1 (CK), 2 (HcT1), 5 (HcT2), and 8 (Jr, 0d-water); groups 1, 3, 6, and 9 (0d-root); groups 1, 4, 7, and 10 (0d-sediment); groups 11, 12, 15, and 18 (30d-water); groups 11, 13, 16, and 19 (30d-root); and groups 11, 14, 17, and 20 (30d-sediment). Moreover, comparisons between different sample days were made: groups 1 and 11; groups 2 and 12; and so on up until groups 10 and 20. The combined groups are as follows: 0 d, group 1, combined groups (2+3+4), (5+6+7), and (8+9+10); 30 d, group 11, (12+13+14), (15+16+17), and (18+19+20); combined groups (2+3+4) and (12+13+14); combined groups (5+6+7) and (15+16+17); combined groups (8+9+10) and (18+19+20). The differences could differ from the phylum to the species level. Comparisons were also performed to identify the microorganisms that can transform in different media (2+5+8 as water, 3+6+9 as root, and 4+7+10 as sediment).

The present study conducted metagenomic testing of the culture water, roots, and sediments to analyze the floating bed (30% and 60% *H. cordata* Thunb; 30% *J. stipulacea*) after 30 days in the presence of tilapia culture water bodies, roots, and bacterial changes at the bottom. To determine the relationship between the prevention of pathogenic bacteria after Chinese medical herb float planting and the transformation of microorganisms, the ratios of beneficial/harmful pathogenic bacteria were calculated [2] and screened within different media [15,16]. It should be considered that environmental factors may shape and affect the diversity of the beneficial/harmful pathogenic bacteria, i.e., a previous study showed that the two most important environmental factors affecting the gut microbiota in diseased fish are the temperature of the water and the ammonia concentration [15]. Moreover, another study on the composition of the farmed *Salmo salar* skin–mucus microbiome before and after netting and transfer showed a limited correlation between the skin–mucus microbiome and the bacterial community present in the rearing water [16]. The analysis for the bacterial community with different groups at 0 and 30 d in the present study was performed without considering the impact of other environmental factors based on the fact that there was no water change or oxygenation.

*2.4. Data Analysis*

The relationship between the selected taxonomic group (abundant phyla, genera, classes, orders, or families), the observed OTUs, or the bacterial community index (Chao1) was calculated using SPSS software (version 27.0). For all parameters, data were compared using a one-way analysis of variance (ANOVA) at the end of each bioassay. A mean comparison was performed using Fisher's least significant difference test (LSD) and Duncan's multiple range test with a significance level of $p < 0.05$.

## 3. Results

*3.1. OTUs Collection, α- and β-Diversity*

We identified 34,373 clean tags (filtered from the original tags with high quality) and 33,627 functional tags in the control water on day 0 (Table S1). Overall, 98% of the sequence length was distributed in the area from 401 to 450 bp, and the average length of the sequence in the control group at 0 d was 412 bp. A total of 4811 bacterial OTUs were identified (97% similarity cut-off). The total counts of the bacterial OTUs were estimated to be 2490 (0d-CK-water), 2663 (0d-HcT1-water), 3027 (0d-HcT1-root), and 2446 (0d-HcT1-sediment). However, no significant differences were observed between the different treatment groups. We compared the Chao1 indices and Shannon diversity indices of the bacterial communities collected from the water, root, and sediment samples. The Chao1 index was determined to be $4916.05 \pm 183.31$, while the Simpson and Shannon indices were $7.94 \pm 1.20$ and $0.968 \pm 0.046$ for the 0d-CK-water group, respectively (Table S1). All the evaluated parameters showed no significant differences among the different groups ($p > 0.05$), implying that the floating planting with different herbs did not affect the species richness and diversity of the microbial communities in the samples.

Proteobacteria, Bacteroidetes, Firmicutes, Actinobacteria, Cyanobacteria, and Fusobacteria were the dominant phyla present in all the samples (Figure S1a). Indeed, the different media and time explained the largest fraction of the variation in β-diversity measured by both weighted and unweighted UniFrac matrices (12.71% along PC1, Figure S1b, $p < 0.05$) after accounting for the variation present, while 5.12% and 4.15% of the variation were explained by PC2 and PC3, respectively, accounting for temperature, light, or other causes. The distributed phyla are shown in Figure S1c, and the biomarkers are shown in Figure S2.

*3.2. Specific OTU Abundances with Chinese Herbs Planting*

With respect to the water samples, 357 shared OTUs were found (Figure S3a); the sensitive phylum shown in Figure S3b, Prevotellaceae, Bacteroidales_S24-7_group, and Lachnospiraceae significantly increased in the CK group; HcT1 and HcT2 increased the levels of Mycobacteriaceae, NS11-12_marine_group, Chitinophagaceae, Saprospiraceae, Lactobacillaceae, and Comamonadaceae, whereas lower rates of *H. cordata* Thunb, Bdellovibrionaceae, Methylophilaceae, and Sphingomonadaceae increased, and the Jr of the *J. stipulacea* group, Comamonadaceae, Ruminococcaceae, and Cytophagaceae increased (Figure S3c). The pathways of flavone and flavonol biosynthesis, galactose metabolism, sphingolipid metabolism, glycosphingolipid biosynthesis-globo series, other glycan degradation, drug metabolism and other enzymes, NOD-like receptor signaling, zeatin biosynthesis, and phenylpropanoid biosynthesis decreased while planting with herbs (Figure S3d).

For the water samples (0 vs. 30 d), the Spirochaetae, Nitrospirae, and Elusimicrobia populations decreased in the CK, HcT1, HcT2, and Jr groups (Figure S4), whereas for the root samples in the CK group, Nitrospirae and Fibrobacteres increased and Chlamydiae decreased. With 60% *H. cordata* Thunb planted under 30 d, Tenericutes, Chlorobi, and Nitrospirae significantly increased, while with lower densities, Fusobacteria and Fibrobacteres increased, and Firmicutes, Chlorobi, and Tenericutes decreased. For the Jr groups, Bacteroidetes, Firmicutes, Fusobacteria, and Nitrospirae decreased, while Actinobacteria, Gemmatimonadetes, Cyanobacteria, and Acidobacteria increased. In the sediment samples, Proteobacteria (except for Jr), Bacteroidetes, Firmicutes, and Fusobacteria decreased, especially in Jr; Chlorobi and Spirochaetae increased in CK; Actinobacteria, Nitrospirae, TM6,

Tenericutes, and Chlamydiae increased in HcT1; Fusobacteria and Fibrobacteres increased in HcT2; and most phyla increased in Jr, especially Cyanobacteria, Gemmatimonadetes, and Acidobacteria.

### 3.3. Biomarker Bacteria with Different Planting Area

The comparison drawn indicated that planting with *H. cordata* Thunb decreased Tenericutes and Deferribacteres while increasing Proteobacteria, Actinobacteria, Gemmatimonadetes, Chlorobi, Spirochaetae, and Nitrospirae. With a larger planting area of *H. cordata* Thunb, Proteobacteria, Chlorobi, Nitrospirae, Gemmatimonadetes, and Spirochaetae increased, while with fewer areas, most phyla decreased except for Actinobacteria and Cyanobacteria. With the same planting area, Jr increased, while Chlorobi, Gemmatimonadetes, Actinobacteria, Fusobacteria, and Fibrobacteres decreased. The proteobacteria and Elusimicrobia populations increased and decreased when the herbs were planted, respectively. However, with a 30% planting area, clavulanic acid biosynthesis, cytochrome p450, apoptosis, the p53 signaling pathway, influenza A, viral myocarditis, small cell lung cancer, colorectal cancer, toxoplasmosis, amyotrophic lateral sclerosis (ALS), pathways in cancer, caffeine metabolism, glycan binding proteins increased, and other pathways were enhanced while planting with herbs (Figure S3d).

### 3.4. Biomarker Bacteria with Different Media Possibly Related to Transformation

In the water control samples, the abundance of Deferribacteres was lower at 30 d when compared to 0 d (Figure S5), which can also be observed in the HcT1 and HcT2 (Armatimonadetes) groups. In the roots and sediment, Tenericutes (both) and Chlamydiae (slightly in the sediment) increased in the HcT1 groups. The abundance of Elusimicrobia and Spirochaetae decreased in the HcT2 root and sediment groups. Tenericutes, WCHB1-60, Elusimicrobia, and Narospirae decreased in the Jr water samples; Bacteroidetes, Firmicutes, and Tenericutes (for roots only) decreased in the root and sediment samples; and most phyla increased in the Jr root and sediment samples.

### 3.5. Beneficial and Harmful Bacteria

The isolated beneficial and harmful bacteria in the comparison among the different groups are revealed in Figure 2. The beneficial bacteria, like Acetobacteraceae, *Roseomonas*, Methylobacteraceae, *Meganema*, and Methylobacterium, increased in the herb groups. Some harmful bacteria, like *Lachnospiraceae*, *Streptococcus danieliae*, *Pseudobutyrivibrio*, and *Lactococcus*, decreased in the herb groups. We combined the samples in the CK, HcT1, HcT2, and Jr groups, with the results showing Armatimonadetes, Deferribacteres, and Fibrobacteres as sensitive biomarkers (Figure 3); the related enriched KEGG pathways are shown in Figure S6.

In the water comparison groups at 30 d, 30% *H. cordata* Thunb increased the pathways of metabolism, human disease, and environmental information processing of the water bacteria, while *H. cordata* Thunb increased the above pathways of the root bacteria, and Jr increased the human disease pathway of the sediment bacteria (Figure 4). The enhanced affected pathways resulting from *H. cordata* Thunb may be higher when compared with Jr. To conclude, the pathways of infectious diseases and environmental acclimation are enhanced within 30 d of herb planting.

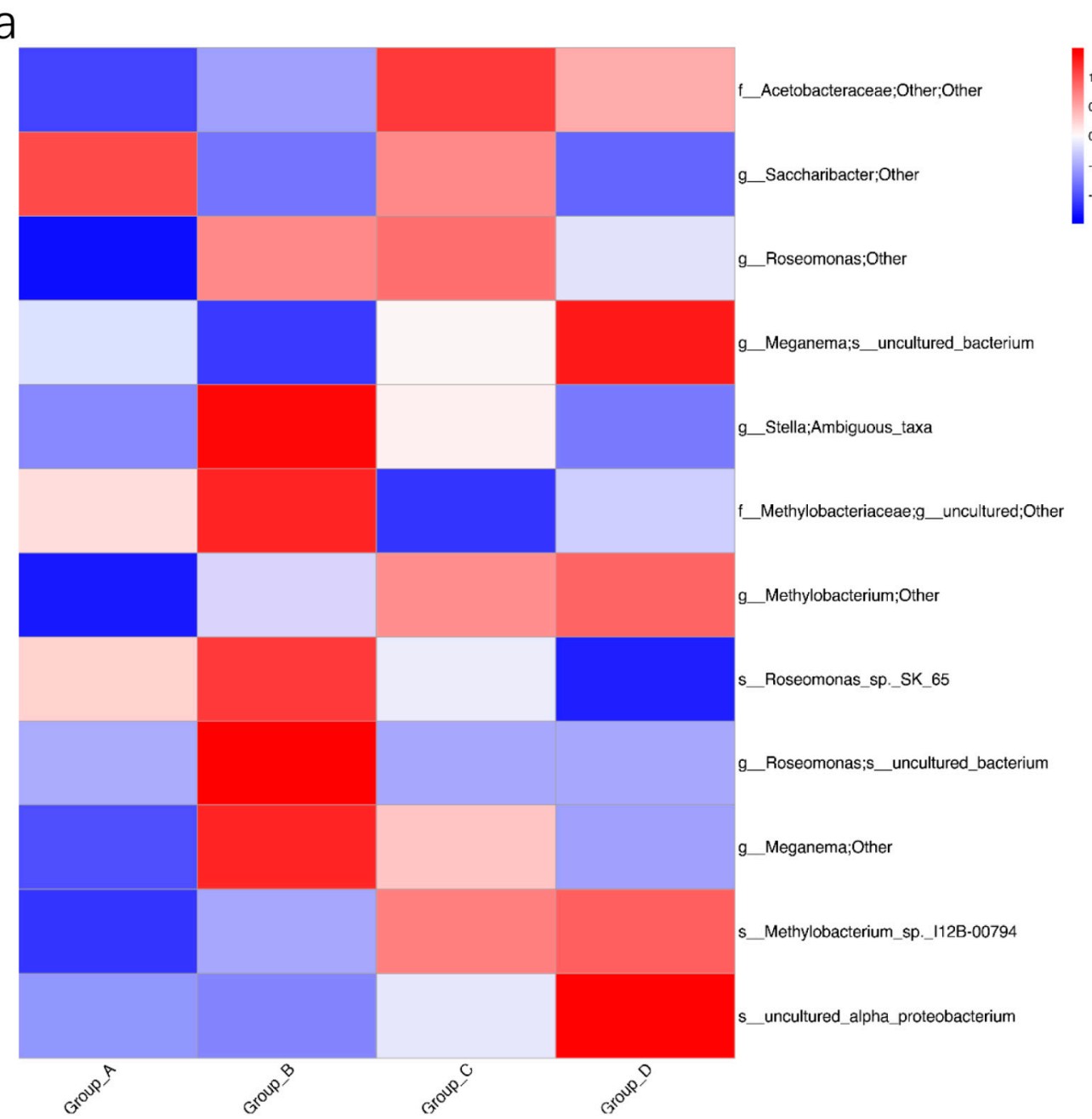

**Figure 2.** *Cont.*

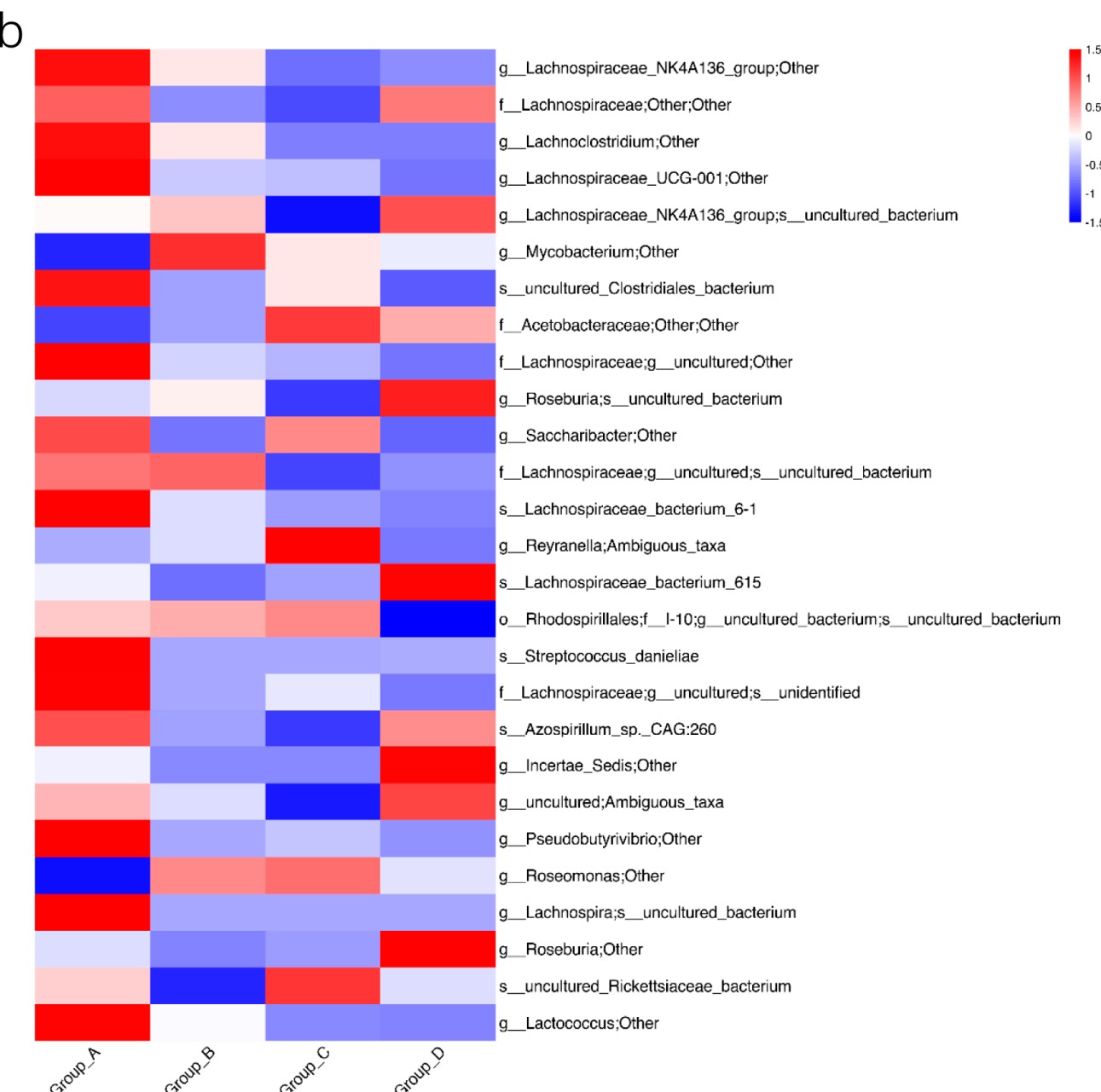

**Figure 2.** The isolated beneficial and harmful bacteria in the comparison among different groups. (**a**) the sensitive beneficial biomarkers, (**b**) the sensitive harmful biomarkers.

Cladogram

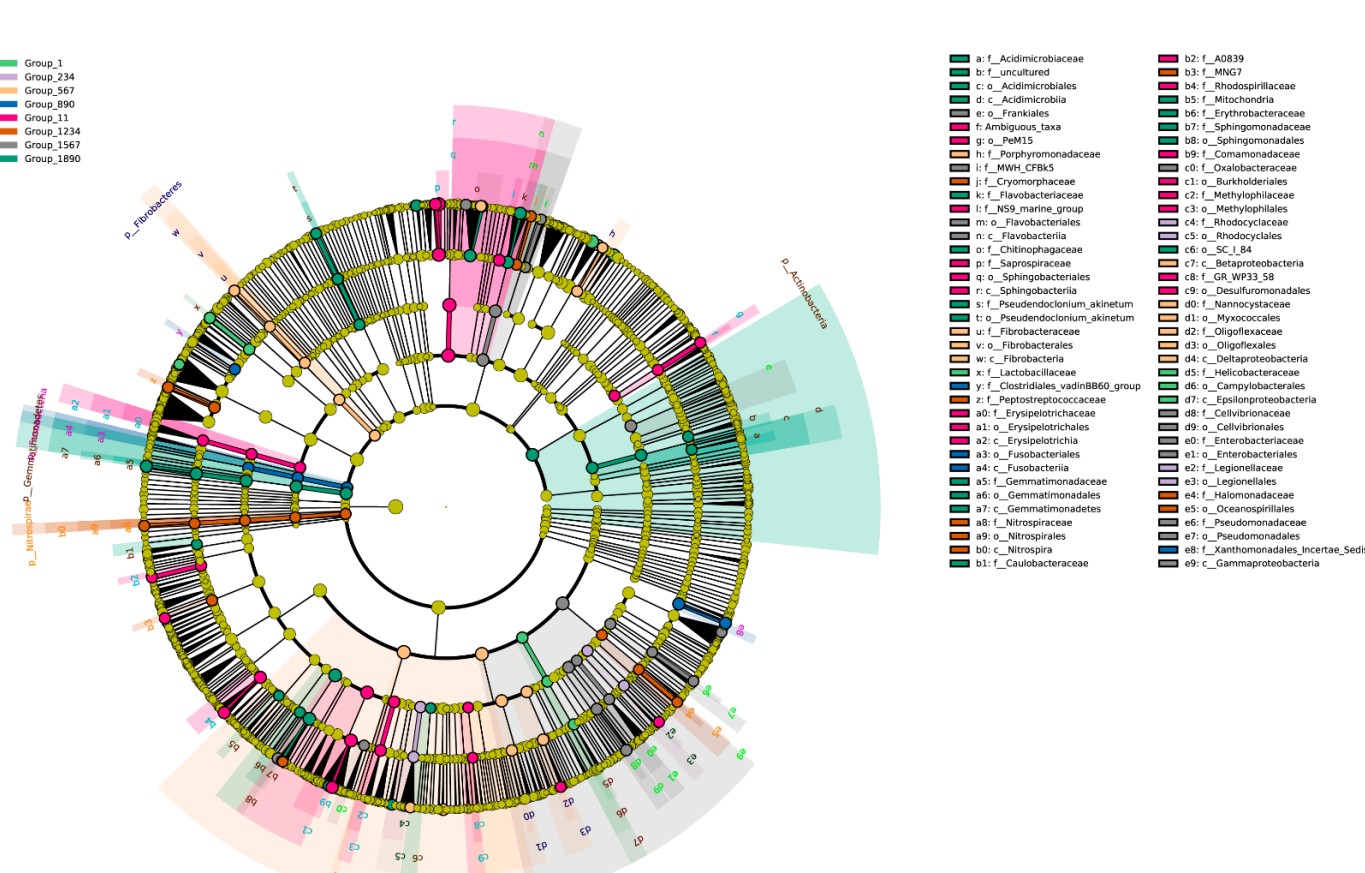

**Figure 3.** The biomarkers among the CK, HcT1, HcT2 and Jr groups at 0 and 30 d found via LefSe (linear discriminant analysis coupled with effect size measurements). CK at 0 and 30 d was named groups 1 and 11, HcT1 at 0 and 30 d was named groups 234 and 1234, HcT1 at 0 and 30 d was named groups 567 and 1567, and Jr at 0 and 30 d was named groups 890 and 1890. Different colors represent different groups, like red nodes, which represent species with relatively high abundance differences in the red group, while yellow nodes represent species with no significant difference in comparison between the two groups. The diameter of nodes is proportional to the relative abundance size. Each layer of nodes represents a phylum/class/order/family/genus from the inside out. The species names represented by English letters in the figure are shown in the legend on the right.

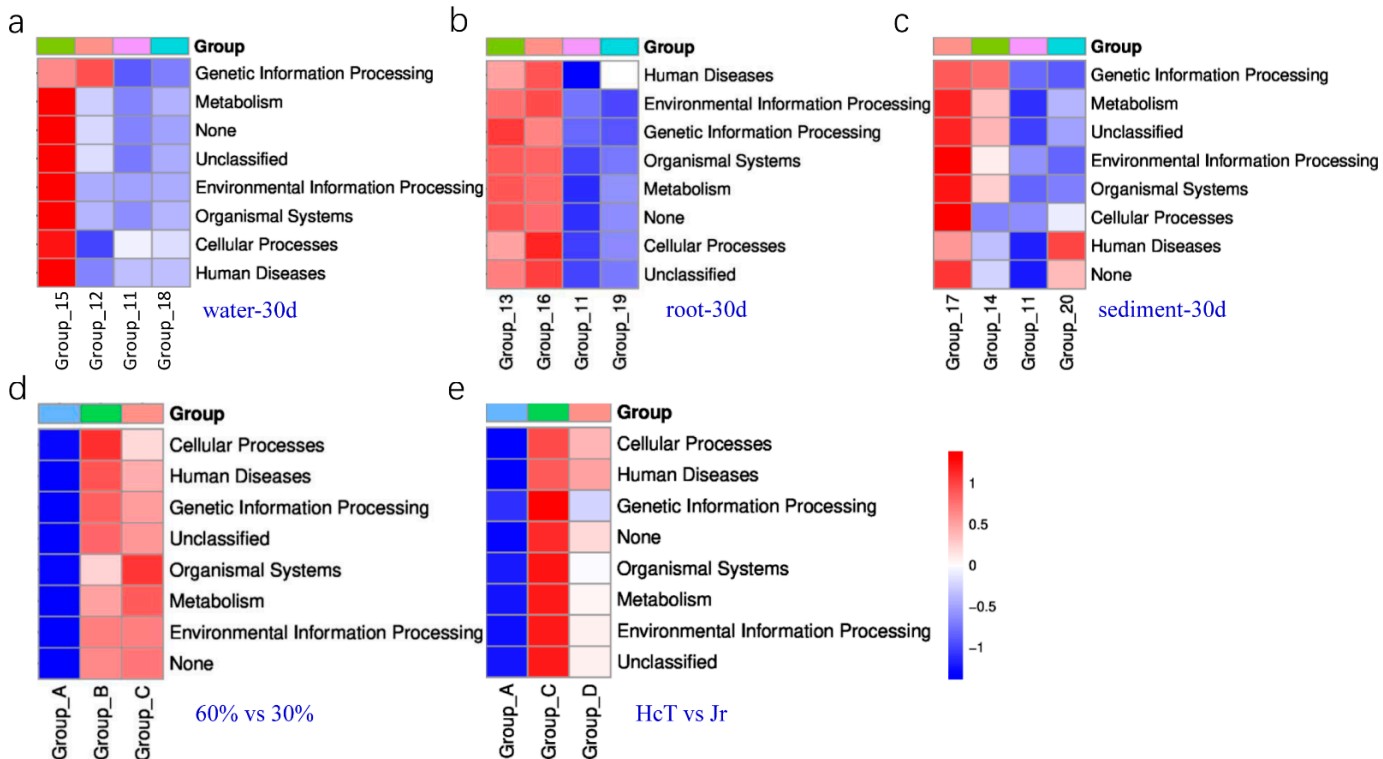

**Figure 4.** The enriched KEGG pathways of the combined samples in different media and groups. Comparisons were also performed to identify the microorganisms that can transform in different media at 30 d (water, roots, and sediment). The 60% HcT combined groups stand for group 2+3+4+12+13+14, while the 30% HcT combined groups stand for group 5+6+7+15+16+17. The HcT combined groups stand for group 2+3+4+12+13+14+5+6+7+15+16+17, while the Jr combined groups stand for group 8+9+10+18+19+20. (**a–c**) stands for the enriched KEGG pathways in the water, root and sediment samples, respectively. (**d**), the pathways involved in the comparison between 60% and 30% planting areas, (**e**), the pathways involved in the comparison between HcT and Jr groups.

## 4. Discussion

Vibrio, Aeromonas, and Shewanella were prevalent in the diseased fish but were rare or even absent in the environmental samples [17]. The dominant phyla were Proteobacteria, Actinobacteria, Cyanobacteria, Bacteroidetes, and Chlorobi in the water samples and Proteobacteria, Firmicutes, Actinobacteria, Chloroflexi, and Bacteroidetes in the sediment samples in the present study. A previous study showed that Proteobacteria and Bacteroidetes were significantly enriched in the soil bacterial community when using microbial inoculants within one month [18], which may demonstrate that planting with herbs may affect the bacterial community or disrupt the crosstalk within 30 days. These findings were partly supported by a prior study that found the dominant phyla to be Firmicutes in sediment samples and Actinobacteria in water samples, with Proteobacteria and Bacteroidetes being dominant in both [14]. The Actinobacteria population increased in the HcT1 sediment groups. The reports showed that high nutrient levels boosted the active metabolism of Actinobacteria when compared to the active metabolism of those in oligotrophic conditions [19], indicating that Actinobacteria may have transformed from water to sediment.

Nitrospirae, which can alleviate nitrite toxicity, are distributed in plant roots, which have an abundance of nitrite-oxidizing bacteria [20]. CyHV-2 abundance is positively correlated with dissolved oxygen levels and the presence of abundant Aeromonas spp. in pond water [21]. Plant roots with stronger dissolved oxygen levels and biological nitrification inhibition capacities suppress the populations of *Nitrospira* in the rhizosphere, which may exert a weak recruitment effect on the soil microbiome [22]. Nitrospirae sig-

nificantly increased and decreased in the HcT1 and Jr root groups, respectively, in the present study, with our previous study results showing that HcT1 and Jr are involved in nitrogen and phosphorus removal. The total phosphorus concentration in the pond sediment was the main reason for cyanobacterial blooms [23], with the Ruminococcaceae, Gemmatimonadetes [24], and Cyanobacteria (sediment) [21] populations increasing in the Jr group [25], with Jr releasing yellow allelochemicals, which may inhibit the growth of algae (like cyanobacterial blooms) [26] and nitrite-oxidizing bacteria (Nitrospirae) [27]; our previous study showed that Jr may have potent algae prevention effects in ponds [4]. However, the abundance of some anti-stress bacteria, such as Chlorobi and Fusobacterium, increased. Floating cultivation with plants will decrease the sulfide concentrations [28], and sulfide also altered the intestinal microbial communities [29]. The abundance of pathogenic bacteria, such as Cyanobacteria, Vibrio, and Photobacterium, increased significantly with exposure to increasing concentrations of sulfide. Microbacterium, Parachlamydia, and Shewanella were all commonly found and downregulated in both sulfide groups, which is associated with adaptation to sulfide stimulation [29].

To explore the effect of different proportions of *J. stipulacea* and *H. cordata* Thunb on the bacterial composition in water samples, roots, and sediments, and in each set of data obtained, the comparison components were divided into three parts, and the influence of Chinese medicinal herbs on the bacterial community was discussed to understand which parts of such herbs can be added to provide the maximum medicinal benefits and their most efficient concentrations [2,3]. In the present study, Chlamydiae decreased in the CK root groups and increased in the plant bioactive compound addition intestinal group [2,3]. Deferribacteres decreased in the HcT1 group, especially in the water samples, which was demonstrated under oligotrophic [30] and anaerobic [31] conditions. The isolated beneficial bacteria Acetobacteraceae has been demonstrated to have an important role in fermentation [32], and it increased in this study, while *Lachnospiraceae* was seen to decrease in the present study, which may be correlative with antibacterial activities [33]. Future applications will be further prospected based on the enriched and enhanced KEGG pathways of infectious diseases, metabolism, and environmental adaptation [2,3]. *H. cordata* Thunb volatile oil inhibits the growth of *Staphylococcus aureus*, *Escherichia coli*, and *Aeromonas hydrophila*. The ethanol extract of *H. cordata* Thunb inhibits the growth of *E. coli*, *A. hydrophila*, and *V. parahaemolyticus*. *J. stipulacea* had a significant inhibitory effect on *Monoraphidium neglectum*. Our previous study demonstrated that planting Chinese herbs enhances intestinal immunity and water nutrient removal rates, releasing allelochemicals against pathogenic bacteria and for algae prevention [4], but the mode of action for this has not been determined yet. Our methods rely on reference genomes in databases that may not always match what is in the sampled ecosystem. Even metagenomic data identifying the presence/absence of bacterial genes do not guarantee functionality in in vivo conditions. The cause of the change in the bacterial community composition of culture water, plant root, and sediment and the mechanism for the potential reduction in the invasion of pathogenic bacteria need to be found.

## 5. Conclusions

The mechanism of the crosstalk exists between the planting root, water, and sediment in the pond has not been determined before this study. This study was conducted on the bacterial composition in water, roots, and sediments on samples of floating beds after 30 days in tilapia field ponds. The results showed that Spirochaetae, Nitrospirae, and Elusimicrobia decreased in water, Tenericutes, Chlorobi, and Nitrospirae increased in the HcT1 roots; and Firmicutes, Fusobacteria decreased in the HcT2 and Jr roots. Actinobacteria, Nitrospirae, Tenericutes, and Chlamydiae increased in the HcT1 sediment; Fusobacteria and Fibrobacteres increased in the HcT2 sediment; and Cyanobacteria, Gemmatimonadetes, and Acidobacteria increased in the Jr sediment. *H. cordata* Thunb decreased Tenericutes and Deferribacteres, while Chlorobi, Nitrospirae, and Gemmatimonadetes increased with a larger planting area, whereas Actinobacteria and Cyanobacteria increased with fewer plant-

ing areas, and Jr only increased Fusobacteria and Fibrobacteres. To conclude, the amount of Proteobacteria increased, while the amount of Deferribacteres and Elusimicrobia decreased when planting with herbs. *H. cordata* Thunb increased the pathways of metabolism, human disease, and environmental information processing of the water and root bacteria, while Jr increased the human disease pathway of the sediment bacteria. The pathways of infectious diseases and environmental acclimation are enhanced within 30 d of herb planting.

**Supplementary Materials:** The following supporting information can be downloaded at https://www.mdpi.com/article/10.3390/environments10120200/s1. Figure S1. The most top 15 phylum (b), PCA analysis (c) and heatmap of phylum via ANOVA (d) among different samples. b, The horizontal and vertical axes represent two characteristic values that can best reflect the variance. c, Horizontal refers to sample information; vertical is species annotation information. Red indicates that the relative abundance of species is high, and blue indicates that the relative abundance of species is low. Figure S2. The species evolution tree and OTU abundance map of top 50 OTUs among different samples. Figure S3. The venn chart (a), heatmap of phylum (b) via ANOVA (c) between 0 and 30 d water samples. Orange represents species with high relative abundance, and blue represents species with low relative abundance from b to d. Figure S4. The top 15 phylum among different comparisons. The above figure shows the abundance changes of species in different groups. Bright green indicates low relative abundance of species, while red indicates high relative abundance of species. Group A, B, C, D stand for control, 60% (HcT1), 30% (HcT2) and 30% (Jr), respectively. Figure S5. The top 50 OTUs found in the water, root and sediment among different comparisons. Figure S6. The significant KEGG pathways among the groups. CK at 0 and 30 d named as group 1 and 11, HcT1 at 0 and 30 d named as group 234 and 1234, HcT1 at 0 and 30 d named as group 567 and 1567, and Jr at 0 and 30 d named as group 890 and 1890. Table S1. Sequencing of the overall situation and $\alpha$-diversity ($n = 3$). Note: clean tags means the filtered ones from the original tags with high quality, and "Avg len (average length)" stands for average length of the OTUs. The Chao index is an estimate of the actual number of OTUs in the community. It is obtained by calculating the number of OTUs detected only once and twice in the community. The value is equal to the estimated number of OTUs. Shannon Wiener index is the Shannon index. The larger the value, the higher the sample diversity and the more uniform the individual distribution. Simpson index refers to the probability that two individuals randomly selected belong to different species. The higher the Simpson index, the higher the community diversity.

**Author Contributions:** Y.Z. and G.X. conceived and designed the experiments; Y.Z. analyzed the data; J.H. contributed reagents/materials/analysis tools; Y.Z. contributed to the preparation of the figures; Y.Z. prepared and wrote the manuscript. All authors have read and agreed to the published version of the manuscript.

**Funding:** The work was supported by the China Agriculture Research System of MOF and MARA (No. CARS-46) and Central Public-interest Scientific Institution Basal Research Fund, CAFS(2023TD66).

**Data Availability Statement:** No new data were created or analyzed in this study. Data sharing is not applicable to this article.

**Acknowledgments:** We thank Ampeire Yona, Noa Shapumba for providing the grammar and spelling check of the manuscript.

**Conflicts of Interest:** The authors declare no conflict of interest.

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
