# Peer review of "Transferred Bacterial Community on the Potentially Pathogenic Bacteria among Aquatic Water, Plant Root, and Sediment When Planting with Chinese Herbs"

_environments, doi:10.3390/environments10120200_

Round 1

Reviewer 1 Report

Comments and Suggestions for Authors

introduction

many papers deal with the description of microbial communities in different vegetation environments, both in terrestrial and aquatic ecosystems. I feel the introduction did no explain sufficient iently what was found in terms of : contrasts between microbes according to the plant species, ans its significance, and 2 ) whether plants that are known to produce secondary compounds are devoted of certain microbes. The bibliography about allelopathy may be helpful. 

methods: for understanding the methods, one may read on the quoted papers. It is not possible. the method should be understandable without making additional bibliography. some abbreviations are not explained, making the understanding impossible (eg "while 60% (HcT1) and 30% (HcT2) ", J. stipulacea (Jr, 30%)...the meaning is obscure). more generally, the fist paragraph is particularly obscure, ad the title of the paragraph is rather strange : "Animals and sample collection"

the figures are in my opinion too numerous and too complex.The legend does not hemet to understand well what is the message that the authors want to provide. I suggest strongly to diminish they number, and to provide a very few number of figures, appropriately legended (the legend may allow to understand the figure content and message without the text).

For example, the figure 1 is clearly too heavy, the legend of the first image is impossible to read, and the 4 images are strongly redundant. 

lnine 197: CK group ? the group significance has not been explicated in the method section (or too unclear)

line 243: "With increasing time in the water control samples, the abundance of 243 Deferribacteres decreased": as far as I understood, there was only two dates of sampling : 0 and 30 days. So this is not possible. it is only possible to say : the abundance was lower or higher at t30 compared to t0.

line 287-292: should appear in the method section, for justifying the search of these phylum (or in the introduction)

line 296-297: Proteobacteria and Bacteroidetes were significantly en- 296 riched in the soil bacterial community using microbial inoculants within one month 297 [22], : this suggests that the authors inoculated bacteria in the sediment ? it was not indicated in the methods. 

line 302 "The Actinobacteria population increased in HcT1 sediment groups 302 with good water quality. ": suggest thee was different water quality in the replicates, it was not indicated in the method section

line 313-314: "Nitrospirae significantly increased and decreased in the HcT1 and Jr root groups, 313 respectively, in the present study, with our previous study results showing that HcT1 314 and Jr are involved in nitrogen and phosphorus removal in water purification. ": water purification is not the purpose of thee study, and the protocol was not designed so ?

Author Response

Introduction

many papers deal with the description of microbial communities in different vegetation environments, both in terrestrial and aquatic ecosystems. I feel the introduction did not explain sufficiently what was found in terms of: contrasts between microbes according to the plant species, and its significance, and 2) whether plants that are known to produce secondary compounds are devoted of certain microbes. The bibliography about allelopathy may be helpful. 

Response: Thank you for the review’s suggestions. The sentence “Floating bed cultivation with J. stipulacea released eight secondary metabolites (i.e., ursolic acid, kaempferol, luteolin, etc.), which played the roles as the allelochemicals. A previous study revealed that volatile allelochemical had a weak antioxidative ability [9], and berberine has been found as the main allelochemical from the herbs implementing the disease inhibitory effects [10]…Our recent study demonstrated that planting with mint (Mentha haplocalyx Briq) can alleviate the toxicological effects produced by methomyl, shape the intestinal microbiota in tilapia, and strengthen the connection between pond water quality and the fish metabolic parameters [14].” has been added in line 45-49, 52-55.

methods: for understanding the methods, one may read on the quoted papers. It is not possible. the method should be understandable without making additional bibliography. some abbreviations are not explained, making the understanding impossible (eg "while 60% (HcT1) and 30% (HcT2) ", J. stipulacea (Jr, 30%)...the meaning is obscure).

Response: Thank you for the review’s suggestions. “as the abbreviation for group” has been added in line 81-84.

more generally, the first paragraph is particularly obscure, and the title of the paragraph is rather strange : "Animals and sample collection"

Response: Thank you, and “Experiment design” has been changed in line 77.

the figures are in my opinion too numerous and too complex. The legend does not hemet to understand well what is the message that the authors want to provide. I suggest strongly to diminish they number, and to provide a very few number of figures, appropriately legended (the legend may allow to understand the figure content and message without the text). For example, the figure 1 is clearly too heavy, the legend of the first image is impossible to read, and the 4 images are strongly redundant. 

Response: Thank you for the review’s suggestions. The figures has been re-drawn (revealed as Fig. 1 in line 268 and Fig. 3 in line 283) and changed (the previous figures were change as the sup. Figures except for Fig. 2).

line 197: CK group ? the group significance has not been explicated in the method section (or too unclear)

Response: Thank you, “Con” has been changed to “CK” in the whole ms. in line 81, 87, 140, 146, 191.

line 243: "With increasing time in the water control samples, the abundance of Deferribacteres decreased": as far as I understood, there was only two dates of sampling : 0 and 30 days. So this is not possible. it is only possible to say : the abundance was lower or higher at t30 compared to t0.

Response: Thank you, corrected as suggested in line 250.

line 287-292: should appear in the method section, for justifying the search of these phylum (or in the introduction)

Response: Thank you, corrected as suggested in line 121-126.

line 296-297: Proteobacteria and Bacteroidetes were significantly enriched in the soil bacterial community using microbial inoculants within one month [22], : this suggests that the authors inoculated bacteria in the sediment ? it was not indicated in the methods. 

Response: Thank you, “A previous study showed that…” has been added in line 316.

line 302 "The Actinobacteria population increased in HcT1 sediment groups with good water quality. ": suggest thee was different water quality in the replicates, it was not indicated in the method section

Response: Thank you, “with good quality” has been deleted in line 323.

line 313-314: "Nitrospirae significantly increased and decreased in the HcT1 and Jr root groups, respectively, in the present study, with our previous study results showing that HcT1 and Jr are involved in nitrogen and phosphorus removal in water purification. ": water purification is not the purpose of the study, and the protocol was not designed so ?

Response: Thank you, “in water purification” has been deleted in line 335.

Reviewer 2 Report

Comments and Suggestions for Authors

The study of Zheng et al., showed the bacterial composition in water, roots and sediments of floating bed samples and the change within a 30-day cultivation period when planted with Houttuynia cordata Thunb and Jussiaea stipulacea. Depending on the samples, several genera decreased in water as Nitrospirae, which increased in roots during that time. This is only one example described as in other ponds other genera came up or decreased. In total, Proteobacteria increased while Deferrbacteres and Elusimicrobia decreased in connection with herbs.

The study shows interesting insights into microbial aquatic systems with bacterial changes of in- and decrease of various genera with the aim to cultivate with specific herbs being able to block a spread of pathogenic genera. It is a highly detailed and carefully conducted study based on 16sRNA gene sequencing. It has rather a descriptive character related to the different ponds but gives high inputs on possibilities of blocking growth of pathogenic bacteria using specific herbs. Now it would be of great interest, weather pathogenic microorganisms are actually suppressed in their growth. Experimental approaches with artificial contaminants can provide information here. Are there any corresponding trials or results? Thus, it could be proven whether the hypothesis of a restriction of the germination of pathogenic microorganisms is actually accompanied by cultivation of the two plants. However, the overall view of a shift of the microbial ecosystem by the plants presented here shows great insights of aquatic systems.

The manuscript is well written and clearly structured. The experimental design contains conclusive assays and the statistical evaluation of the results proves the statements.

Some remarks:

The last sentence in the abstract and in the conclusion is identical. Please change. (Lines 28 and 371)

Lines 27 (abstract) and line 370 (conclusion): “Elusimicrobia” is listed double. It that how it should be?

Line 13 change “has” to “have”

Line 44: you introduce N and P for the first time. Please write once the appreviation in detailed specification.

Line 149 (figure 1): do you mean PCR instead of PCA analysis?

Line 323: … will decrease…

Line 353: …in in-vivo…

Comments on the Quality of English Language

minor editing

Author Response

The study of Zheng et al., showed the bacterial composition in water, roots and sediments of floating bed samples and the change within a 30-day cultivation period when planted with Houttuynia cordata Thunb and Jussiaea stipulacea. Depending on the samples, several genera decreased in water as Nitrospirae, which increased in roots during that time. This is only one example described as in other ponds other genera came up or decreased. In total, Proteobacteria increased while Deferrbacteres and Elusimicrobia decreased in connection with herbs.

The study shows interesting insights into microbial aquatic systems with bacterial changes of in- and decrease of various genera with the aim to cultivate with specific herbs being able to block a spread of pathogenic genera. It is a highly detailed and carefully conducted study based on 16sRNA gene sequencing. It has rather a descriptive character related to the different ponds but gives high inputs on possibilities of blocking growth of pathogenic bacteria using specific herbs. Now it would be of great interest, weather pathogenic microorganisms are actually suppressed in their growth. Experimental approaches with artificial contaminants can provide information here. Are there any corresponding trials or results?

Response: Thank you for the review’s suggestions. The question of “whether pathogenic microorganisms are actually suppressed in their growth?” needs to be determined, which may be of great interest for public aquatic researcher and fish farmers. This study conducted 16S rRNA gene sequencing on samples of the same to analyze the floating beds (60% and 30% H. cordata Thunb, J. stipulacea named HcT1, HcT2, and Jr) after 30 days in the presence of tilapia culture water, roots, and sediments with bacterial community changes in the respective experimental groups. When planting with herbs, results showed that Proteobacteria increased, while Deferribacteres, Elusimicrobia, and Elusimicrobia decreased, which hinted the pathogenic genera may transfer among the water, plant roots, and sediments. And finally, this study suggested that floating cultivation with herbs may be beneficial for blocking the spread of the pathogenic genera found in the samples. The authors tended to find the corresponding results of the pathogenic microorganisms, but the identification for the pathogenic genera needed the sequence data from the environments in aquatic ponds (water, sediments, fish gut, gill, etc,). The revised version added the isolated beneficial and harmful bacteria (revealed in Fig. 2) and the details for the comparisons for different media, planting area, and herbs (shown in Fig. 3). The corresponding result and discussion has also been added in line 258-262, 290-295, 362-365. The further study for isolating the aquatic pathogenic genera will be performed, based on this study.

Thus, it could be proven whether the hypothesis of a restriction of the germination of pathogenic microorganisms is actually accompanied by cultivation of the two plants. However, the overall view of a shift of the microbial ecosystem by the plants presented here shows great insights of aquatic systems.

The manuscript is well written and clearly structured. The experimental design contains conclusive assays and the statistical evaluation of the results proves the statements.

Response: Thank you for the review’s suggestions.

Some remarks:

The last sentence in the abstract and in the conclusion is identical. Please change. (Lines 28 and 371)

Response: Thank you, we maintained the sentence in the abstract, while changed it in the conclusion in line 396-402.

Lines 27 (abstract) and line 370 (conclusion): “Elusimicrobia” is listed double. It that how it should be?

Response: Thank you, “Elusimicrobia” has been deleted in line 28, 395.

Line 13 change “has” to “have”

Response: Thank you, corrected as suggested in line 13.

Line 44: you introduce N and P for the first time. Please write once the appreviation in detailed specification.

Response: Thank you, corrected as suggested in line 44-45.

Line 149 (figure 1): do you mean PCR instead of PCA analysis?

Response: Thank you, PCA analysis rotated the original data space such that the axes of the new coordinate system point into the directions of highest variance of the data. PCR possibley stands for “Polymerase Chain Reaction”. We changed figure 1 as the sup. figure.

Line 323: … will decrease…

Response: corrected as suggested in line 343.

Line 353: …in in-vivo…

Response: corrected as suggested in line 377.

Reviewer 3 Report

Comments and Suggestions for Authors

After reading the manuscript under the title Transferred bacterial community on the potentially pathogenic bacteria among aquatic water, plant root, and sediment when planting with Houttuynia cordata Thunb and Jussiaea stipulacea one can suggest that it should be rewritten (as well as its title that do not correspond completely to the text) since, in its present state, it is not showing major concept where the authors hypothesised on possible beneficial influence of plants tested and on the possible prevention of pathogenic bacteria (and the transformation in microbial populations) after mentioned medical herbs (H. cordata Thunb and J. stipulacea) that have been grown in floating beds over the fish tanks. Moreover, the authors found increased activity in markers of metabolism pathways, some human diseases, environmental information processing of the water and plant root bacteria.

 I found the manuscript well-organized, and it contains all expected paragraphs. According to usual rules and the expectation that someone interested will want to reproduce the described research,  I'm of the opinion that section Material and Methods should be developed and supported by detailed with clear descriptions/explanations of all methods and material that was performed/used. Even some basic information (e.g., on organisms - Salmo salar is mentioned later in 2.2. and some other important experimental facts one can read not earlier than in Discussion).

The hypothesis for this study was that the pathogenic bacterial genera may transfer among the water, plant roots, and sediments, and floating cultivation with herbs may be beneficial for blocking the spread of the pathogenic genera found in the samples (as it is mentioned at the end of Introduction. It is not clear (from all that was written in Discussion) if results obtained are sufficient if the authors in the Conclusion emphasized that mechanism of the crosstalk exists has not been determined.

Maybe it is all about inconsistency in writing, but in every case this manuscript should be thoroughly re-read and re-processed to become cleaner and more grounded on the results that were obtained (that are interesting and very informative, but they should be presented in more understandable way). Special effort should be invested in improvement of the main title of the manuscript, and it should accentuate major aspects and achievements of the research.

There were no major problems concerning literature used.

 Minor remarks:

Citation mentioned near the left border of the text should be omitted since it is not connected with the manuscript: Yin T.T.; Zheng Y.; Liu T.Y.; Wang X.F.; Gao J.C.; Nie Z.J.; Song L.L.; Xu G.C.; Yuan J.L. Study on water purification effect and operation parameters of various units of wastewater circulation . Water 2022, 14, x. https://doi.org/10.3390/xxxxx...

 Some other minor comments were marked and elaborated directly in the manuscript (comments and marks)  the file is attached.

 In its present form the manuscript is not suitable for publication in the Environments. If it will be reconsider after major changes it will get another peer review round, but before that it will have to be corrected thoroughly by the authors.

Author Response

After reading the manuscript under the title “Transferred bacterial community on the potentially pathogenic bacteria among aquatic water, plant root, and sediment when planting with Houttuynia cordata Thunb and Jussiaea stipulacea” one can suggest that it should be rewritten (as well as its title that do not correspond completely to the text) since, in its present state, it is not showing major concept where the authors hypothesised on possible beneficial influence of plants tested and on the possible prevention of pathogenic bacteria (and the transformation in microbial populations) after mentioned medical herbs (H. cordata Thunb and J. stipulacea) that have been grown in floating beds over the fish tanks. Moreover, the authors found increased activity in markers of metabolism pathways, some human diseases, environmental information processing of the water and plant root bacteria.

Response: Thank you for the review’s suggestions. Our previous studies showed that Houttuynia cordata Thunb floating beds can significantly improve non-specific immunity in tilapia, and Chinese herbs served as an important substitute for antibiotic for disease prevention. While the crosstalk between the planting root, water, and sediment after planting has not been determined before. The present study showed the pathogenic genera may transfer among the water, plant roots, and sediments, and floating cultivation with herbs may be beneficial for blocking the spread of the pathogenic genera found in the samples. The authors has revised some sentence to keep in line with our goal (i.e. line 261). And especially the title for each part has been revised in line 116-117, 206, 247, 261, 270.

 I found the manuscript well-organized, and it contains all expected paragraphs. According to usual rules and the expectation that someone interested will want to reproduce the described research,  I'm of the opinion that section Material and Methods should be developed and supported by detailed with clear descriptions/explanations of all methods and material that was performed/used. Even some basic information (e.g., on organisms - Salmo salar is mentioned later in 2.2. and some other important experimental facts one can read not earlier than in Discussion).

Response: Thank you for the review’s suggestions. The details in the section Material and Methods have been elaborated (i.e. line 85, 93-97, etc.). About the Salmo salar, another reviewer said “the previous Ms. line 287-292: should appear in the method section, for justifying the search of these phylum (or in the introduction)”, and the authors moved this to the section Material and Methods as the response, which may make the potential readers confused. The sentence “and screened within different medias [22-23]. Considering the environmental factors may shape and affect the diversity of the beneficial/harmful pathogenic bacteria, i.e., a previous study showed that t…nd another…The analysis for the bacterial community with different groups at 0 and 30 d in the present study has been performed without considering the impact of other environmental factors, based on the fact of no water change and oxygenation.” has been added in line 139-141, 146-149.

The hypothesis for this study was that the pathogenic bacterial genera may transfer among the water, plant roots, and sediments, and floating cultivation with herbs may be beneficial for blocking the spread of the pathogenic genera found in the samples (as it is mentioned at the end of Introduction. It is not clear (from all that was written in Discussion) if results obtained are sufficient if the authors in the Conclusion emphasized that “mechanism of the crosstalk exists has not been determined”.

Response: Thank you for the review’s suggestions. The authors acknowledged the conclusion was the summary needing the background. This sentence in the Discussion has been revised (“before this study” added in line 387).

Maybe it is all about inconsistency in writing, but in every case this manuscript should be thoroughly re-read and re-processed to become cleaner and more grounded on the results that were obtained (that are interesting and very informative, but they should be presented in more understandable way). Special effort should be invested in improvement of the main title of the manuscript, and it should accentuate major aspects and achievements of the research.

Response: Thank you for the review’s suggestions. The details (title and presentation for the results) has been added and revised in line 227-232 and 256-260; line 116-117, 206, 247, 261, 270.

There were no major problems concerning literature used. Minor remarks:

Citation mentioned near the left border of the text should be omitted since it is not connected with the manuscript: “Yin T.T.; Zheng Y.; Liu T.Y.; Wang X.F.; Gao J.C.; Nie Z.J.; Song L.L.; Xu G.C.; Yuan J.L. Study on water purification effect and operation parameters of various units of wastewater circulation . Water 2022, 14, x. https://doi.org/10.3390/xxxxx”...

Response: Thank you, and the reference has been changed in line 450-455.

 Some other minor comments were marked and elaborated directly in the manuscript (comments and marks) – the file is attached.

Response: Thank you, corrected as suggested (i.e. line 16-17, etc.).

In its present form the manuscript is not suitable for publication in the Environments. If it will be reconsider after major changes it will get another peer review round, but before that it will have to be corrected thoroughly by the authors.

Response: Thank you for the review’s suggestions. The revised Ms. after reviewing by all the authors has been attached for re-consideration.

Reviewer 4 Report

Comments and Suggestions for Authors

General comment: consider providing specific objectives and objectives of this work.

Line 46 – either rephrase the sentence to make a general statement related to secondary metabolites or also provide the remaining one for the understanding of readers.

Line 48- The authors should provide what is berberine and where it’s found within plants or in which environment.

Line 51 – but methomyl is an insecticide, so how is it linked with nutrient absorption?

Line 52 – correct as “tilapia, strengthen the connection between……..quality, and …….”

Line 78 – If possible, authors should provide a picture of the setup used for the experiment, seeing such images will give the reader a much clearer view of the experiment performed.

Line 140 – I did not manage to find the supplementary information related to figure S1A provided. Authors must make sure that this is available if the manuscript is promoted for publication. I guess it provided by the formatting is very unsatisfactory. Authors must group them so that the figures can be traced easily.

Line 164 – do not the authors think that the caption provided for figure S1 is unnecessarily long? I suggest that this should be summarised, and the details provided here can be moved to the place where the authors are explaining them in the text.

Line 220 – to my understanding authors have not provided the details of this analysis in the materials and methods.

Line 227 – I suggest that authors should use a more reading-friendly treatment nomenclature for the abbreviation of the treatments (Like T1 = CK, T2 = HcT1, and T3 = HcT2, and T4 = Jr), instead of CK, HcT1, HcT2, and Jr. It is very confusing for me to recall these.

Line 250 – What could be the reason for the decrease in Deferribacteres abundance?

Line 265 – Information for KEGG pathway analysis is missing in methods.

Line 270- In Figure 2 the taxonomic cladogram information for nodes is missing. I think due to the inability of the data presentation authors should only provide the information for nodes that have relatively high abundance differences, or vice versa.

Comments on the Quality of English Language

The manuscript is written in readable English quality with minor revisions needed

Author Response

Line 46 – either rephrase the sentence to make a general statement related to secondary metabolites or also provide the remaining one for the understanding of readers.

Response: Thank you for the review’s suggestions. “bioactive compounds….. for harmful bacterial prevention” has been added in line 46-48.

Line 48- The authors should provide what is berberine and where it’s found within plants or in which environment.

Response: Thank you for the review’s suggestions. The sentence “(a kind of alkaloid extracted from the plants)…root of Coptis chinensis” has been added in line 49-50.

Line 51 – but methomyl is an insecticide, so how is it linked with nutrient absorption?

Response: Thank you for the review’s suggestions. The sentence “such aquatic plants have the advantage of nutrient absorption” means “some vegetable-like plants, like water spinach…”, not methomyl in line 51-53.

Line 52 – correct as “tilapia, strengthen the connection between……..quality, and …….”

Response: Thank you, corrected as suggested in line 55.

Line 78 – If possible, authors should provide a picture of the setup used for the experiment, seeing such images will give the reader a much clearer view of the experiment performed.

Response: Thank you for the review’s suggestions, and Fig.1 has been added in line 98.

Line 140 – I did not manage to find the supplementary information related to figure S1A provided. Authors must make sure that this is available if the manuscript is promoted for publication. I guess it provided by the formatting is very unsatisfactory. Authors must group them so that the figures can be traced easily.

Response: Thank you for the review’s suggestions, and Figure S1a has been deleted.

Line 164 – do not the authors think that the caption provided for figure S1 is unnecessarily long? I suggest that this should be summarised, and the details provided here can be moved to the place where the authors are explaining them in the text.

Response: Thank you for the review’s suggestions, and some sentences have been deleted in line 188-196.

Line 220 – to my understanding authors have not provided the details of this analysis in the materials and methods.

Response: Thank you for the review’s suggestions. The sentence “, and the relative enriched KEGG pathways have been simultaneously analyzed [2-4, 13, 21]” has been added in line 121-122.

Line 227 – I suggest that authors should use a more reading-friendly treatment nomenclature for the abbreviation of the treatments (Like T1 = CK, T2 = HcT1, and T3 = HcT2, and T4 = Jr), instead of CK, HcT1, HcT2, and Jr. It is very confusing for me to recall these.

Response: Thank you for the review’s suggestions. The authors do not use T1-T4, because Fig. 1 has been added to make the potential readers more clearly.

Line 250 – What could be the reason for the decrease in Deferribacteres abundance?

Response: Thank you for the review’s suggestions. A previous study showed a band related to Deferribacteres was enhanced in eastern sites with a natural gradient of strong P limitation in Florida Bay (Guevara et al., 2014). The sentence “Deferribacteres decreased in the HcT1 group, especially in water samples, which was demonstrated under oligotrophic [38–39] and anaerobic [40] conditions” revealed in line 355-357. Considering the method for this study was without oxygenation, or even nutrient exchange because we used glass tank.

Reference:

Guevara R, Ikenaga M, Dean AL, Pisani C, Boyer JN. Changes in sediment bacterial community in response to long-term nutrient enrichment in a subtropical seagrass-dominated estuary. Microb Ecol. 2014,68(3):427-40. doi: 10.1007/s00248-014-0418-1.

Line 265 – Information for KEGG pathway analysis is missing in methods.

Response: Thank you for the review’s suggestions. The sentence “, and the relative enriched KEGG pathways have been simultaneously analyzed [2-4, 13, 21]” has been added in line 121-122.

Line 270- In Figure 2 the taxonomic cladogram information for nodes is missing. I think due to the inability of the data presentation authors should only provide the information for nodes that have relatively high abundance differences, or vice versa.

Response: Thank you for the review’s suggestions. Figure 3 (previous Fig. 2) has been replaced with more clear information and nodes in line 281.

Round 2

Reviewer 3 Report

Comments and Suggestions for Authors

.
The manuscript, the title and many subtitles have been rewritten mostly according to suggested  reviewers remarks from 1st round of review. At present the readers will easily take important information on research achievements (facts that floating cultivation of mentioned herbs is beneficial for stopping the transfer of the pathogenic bacteria in the system and details on experiments performed). Material and Methods and other sections of the manuscript has been developed extensively as well.

In its present form the manuscript is suitable for publication in the Environments, but some minor changes are suggested as follows:

Line26:
areas = area

L94:
Group1 = Group 1

L95:
Group2 = Group 2

L96:
Group3 = Group 3

L97:
Group4 = Group 4

L244 in subtitle 3.4.:
possiblely  = possibly
-

Author Response

The manuscript, the title and many subtitles have been rewritten mostly according to suggested  reviewer’s remarks from 1st round of review. At present the readers will easily take important information on research achievements (facts that floating cultivation of mentioned herbs is beneficial for stopping the transfer of the pathogenic bacteria in the system and details on experiments performed). Material and Methods and other sections of the manuscript has been developed extensively as well. In its present form the manuscript is suitable for publication in the Environments, but some minor changes are suggested as follows:

Line26:
“areas” = area

Response: Thank you, corrected as suggested.

L94:
“Group1” = Group 1

Response: Thank you, corrected as suggested.

L95:
“Group2” = Group 2

Response: Thank you, corrected as suggested.

L96:
“Group3” = Group 3

Response: Thank you, corrected as suggested.

L97:
“Group4” = Group 4

Response: Thank you, corrected as suggested.

L244 in subtitle 3.4.:
“possiblely”  = possibly

Response: Thank you, corrected as suggested.

Reviewer 4 Report

Comments and Suggestions for Authors

The authors have significantly improved the manuscript. However, some points/minor changes are needed

Line 45: “bioactive compounds (i.e., ursolic acid, kaempferol, luteolin, etc.)” Here it was suggested that authors should either provide the name of all compounds or should just as a general phrase, even though the authors have added the words still it seems incomplete. So it is recommended that the name of all such bioactive compounds should be provided for the ease of readers.

Line 51: Authors must provide the significance of methomyl and nutrient absorption, what this indicates, and how these are associated.

Comments on the Quality of English Language

The authors' work presented here is not found to have any detectable anomalies, however, authors should carefully revisit the whole work to avoid any unnecessary typos or grammatical mistakes.

Author Response

The authors have significantly improved the manuscript. However, some points/minor changes are needed. Line 45: “bioactive compounds (i.e., ursolic acid, kaempferol, luteolin, etc.)” Here it was suggested that authors should either provide the name of all compounds or should just as a general phrase, even though the authors have added the words still it seems incomplete. So it is recommended that the name of all such bioactive compounds should be provided for the ease of readers.

Response: Thank you, corrected as suggested, and “palmitic acid, sitosterol, betulinic acid, gallic acid,…quercetin 3-O-rhamnoside,” has been added in line 45-6.

Line 51: Authors must provide the significance of methomyl and nutrient absorption, what this indicates, and how these are associated.

Response: Thank you, corrected as suggested, and “which can be useful for in-situ remediation with enhancement of water quality and removal of pesticide simultaneously,” has been added in line 51-3.

The authors' work presented here is not found to have any detectable anomalies, however, authors should carefully revisit the whole work to avoid any unnecessary typos or grammatical mistakes.

Response: Thank you, corrected as suggested, i.e. line 26, 246.